# Efficacy and Feasibility of Salvage Re-Irradiation with CyberKnife for In-Field Neck Lymph Node Recurrence: A Retrospective Study

**DOI:** 10.3390/jcm8111911

**Published:** 2019-11-07

**Authors:** Daijiro Kobayashi, Hiro Sato, Jun-ichi Saitoh, Takahiro Oike, Atsushi Nakajima, Shin-ei Noda, Shingo Kato, Mototaro Iwanaga, Tsuneo Shimizu, Takashi Nakano

**Affiliations:** 1CyberKnife Center, Kanto Neurosurgical Hospital, Kumagaya 360-0804, Japan; 2Department of Radiation Oncology, Gunma Prefectural Cancer Center, Ota 373-0828, Japan; 3Department of Radiation Oncology, Gunma University Graduate School of Medicine, Maebashi 371-8511, Japan; 4Department of Radiation Oncology, Toyama University Graduate School of Medicine, Toyama 930-0194, Japan; 5Division of Gene Regulation, Institute for Advanced Medical Research, School of Medicine, Keio University, Shinjuku-ku 160-8582, Japan; 6Department of Radiation Oncology, Saitama Medical University International Medical Center, Hidaka 350-1298, Japan; 7National Institute of Radiological Sciences Hospital, National Institutes for Quantum and Radiological Science and Technology, Inage-ku, Chiba 263-8555, Japan

**Keywords:** salvage treatment, in-field recurrence, re-irradiation, stereotactic radiation therapy, neck lymph node recurrence

## Abstract

Neck lymph node (LN) recurrence in the irradiated field represents an important aspect of treatment failure after primary radiotherapy owing to the lack of a standard treatment. The aim of this study is to investigate the efficacy and safety of CyberKnife treatment for neck LN recurrence after radiotherapy. Between 2008 and 2016, 55 neck LN recurrences after radiotherapy in 16 patients were treated with CyberKnife. The median follow-up period was 17 months (range, 2–53 months). The median previous radiotherapy dose was 68 Gy (range, 50–70 Gy). The median marginal dose as equivalent dose delivered in 2-Gy fractions (α/β = 10) was 50 Gy (range, 40–58 Gy). The one-year local control (LC) and overall survival rates were 81% and 71%, respectively. The one-year LC was higher with a target volume ≤1.0 cm^3^ than that with a target volume >1.0 cm^3^ (*p* = 0.006). Fatal bleeding was observed in one patient who had large (91 cm^3^) and widespread tumor with invasion to the carotid artery before CyberKnife treatment. CyberKnife treatment for neck LN recurrence is safe and feasible in most cases. Indication for the treatment should be carefully considered for large and widespread tumors.

## 1. Introduction

More than 30% of head and neck cancers are diagnosed in advanced stages [1]. The standard treatment for inoperable locally advanced head and neck cancers is chemoradiotherapy (CRT) [2,3,4,5], whose efficacy has been proven by a meta-analysis [6,7]. However, local/regional (locoregional) recurrence after CRT occurs in approximately 15% of patients [8], where the recurrence occurs mainly within the area that received high-dose irradiation [9].

Locoregional recurrence in the head and neck can greatly diminish the quality of life (QOL) of patients with various symptoms including swelling, pain, and pharynx ulceration. More serious reason is that it can cause death from bleeding of ulcerated lesions. Surgical resection as a salvage treatment for locoregional recurrence shows overall survival (OS) rate of 20% at 3 years postoperatively [10]. However, salvage surgery is feasible in only 7–27% of patients because of extensive tumor invasion and a high risk of postoperative complications in the irradiated area [10,11,12]. With regard to lymph node (LN) recurrence, only 20% of patients can be considered as candidates for salvage surgery [12]. Conversely, conventional radiotherapy is unsuitable for in-field recurrence, including LN recurrence, owing to the risk of severe adverse reactions after curative irradiation. Previous studies have reported severe grade 3–5 toxicities in 45–85% of patients and poor survival rates after re-irradiation with three-dimensional conformal radiotherapy (3DCRT) and intensity-modulated radiotherapy (IMRT) techniques [13,14,15,16]. Therefore, chemotherapy alone is commonly performed for locoregional recurrence; however, the median survival after chemotherapy duration is limited to 5–9 months [17,18].

Recent technological advances have made it possible to achieve highly conformal dose distributions to recurrent tumor using stereotactic radiosurgery (SRS) or stereotactic radiation therapy (SRT), resulting in high local control (LC) and low toxicity, with a reduction in the dose to organs at risk (OAR) [19,20,21]. Many studies have reported the efficacy and safety of SRS or SRT using CyberKnife as a salvage re-irradiation approach for local recurrence at the primary site in the head and neck [22,23,24,25,26,27]. However, few reports have mentioned salvage re-irradiation for LN recurrence in the irradiated field. Therefore, the efficacy and safety of re-irradiation with CyberKnife on neck LN recurrence is not fully elucidated yet. To address this issue, we conducted a retrospective study that investigates CyberKnife-treated 55 in-field neck LN lesions recurring after radiotherapy.

## 2. Material and Methods

### 2.1. Patients

We conducted a retrospective chart review of 55 lesions from 16 consecutive patients with in-field neck LN recurrence treated with CyberKnife at our institution between July 2008 and March 2016. All patients provided written informed consent prior to treatment. The study protocol was approved by the Ethics Review Board of the Kanto Neurosurgical Hospital. This trial has been registered in the University Hospital Medical Information Network Clinical Trials Registry (UMIN-CTR; number 000031155).

The eligibility criteria for this analysis were as follows: (a) history of external beam radiotherapy for head and neck cancers as primary treatment or postoperative radiotherapy with or without chemotherapy; (b) in-field neck LN recurrence detected by a combination of approaches, including computed tomography (CT), fluorodeoxyglucose (FDG)-positron emission tomography (PET)/CT, blood tests, and clinical assessments by two or more radiologists. Based on these criteria, a total of 55 lesions from 16 patients were eligible for inclusion in the final analysis. The lymph node recurrence occurred in areas that received high-dose irradiation, and biopsy was not performed as it could lead to refractory skin ulceration.

The patient characteristics are summarized in Table 1. The primary tumor sites were the nasopharynx (17 lesions), oropharynx (three lesions), hypopharynx (five lesions), buccal mucosal (13 lesions), tongue (two lesions), and larynx (15 lesions). All recurrent diseases were considered unresectable by head and neck surgeons. The patient with buccal mucosal tumor received CyberKnife treatment three times. Conversely, one of the two patients with laryngeal tumor received CyberKnife treatment five times whereas the other patient with laryngeal tumor received CyberKnife treatment three times.

All patients had received previous radiotherapy for the neck. Among the patients, 10 (63%) received radiotherapy as primary treatment, 4 (25%) received radiotherapy as adjuvant postoperative therapy, and 2 (12%) received radiotherapy for local or LN recurrence after surgery. Four patients receiving radiotherapy with doses ranging from 50 to 64.8 Gy as adjuvant postoperative therapy harbored recurrence risk factors such as poorly differentiated pathology, metastases in several lymph nodes in the neck, and positive lymph nodes with extranodal extension. In contrast, two patients who received radiotherapy for local or LN recurrence after surgery did not have any recurrence risk factors and thus did not receive adjuvant radiotherapy. One of the two patients had local recurrence nine months after the surgery for tongue tumor and received concurrent chemoradiotherapy at a dose of 65.5 Gy. The other patient had neck lymph node recurrence two months after the surgery for laryngeal tumor and received radiotherapy at a dose of 50 Gy. No patient had evidence of disease at the primary site, and lung metastasis was noted in one patient when starting CyberKnife treatment. The median previous conventional radiation dose was 68 Gy (range, 50–70 Gy). Neck dissection had been performed as primary treatment in five patients (31%) and as secondary treatment in two patients (13%). The median interval between completion of prior radiotherapy and CyberKnife treatment was 25 months (range, 6–69 months). The median target volume was 1.2 cm^3^ (range, 0.05–91 cm^3^). There was no evidence of recurrence at the primary site during CyberKnife treatment.

### 2.2. Treatment

All patients included in this study were treated using CyberKnife G3 and G4 (Accuray, Sunnyvale, CA, USA). All treatment procedures were performed under CT and magnetic resonance image (MRI) guidance in a frameless system. OAR, such as the skin, mucosa, trachea, and brain stem, were identified using CT and MRI. The gross tumor volume (GTV) was delineated on the basis of fusion images of enhanced CT and MRI using 1.0-mm thick axial images. The clinical target volume (CTV) was identical to the GTV (CTV = GTV). No additional margin was added to the planning target volume (PTV; i.e., GTV = CTV = PTV). Coverage of more than 90% of the target volume was set to cover with the 50–70% isodose line. The marginal dose prescription was defined as the percentage (100% = max dose) of the isodose curve covering the PTV. The minimum dose covering 95% (D95) [or the minimum dose covering 90%; (D90)] was defined as the minimum dose covering 95% (or 90%) of the PTV. The biological effective dose (BED) was calculated as follows: BED = nd (1 + d/α/β), where *n* = fraction number, d = fraction dose, α/β = 10 (tumor) or 3 (surrounding normal tissue). Composite BEDs were calculated as 3D dose distributions in equivalent doses delivered in 2-Gy fractions (EQD2). The target marginal dose was 18–20 Gy administered in one fraction. The treatment doses and fractions were determined according to the tumor volume and surrounding critical structures. If the dose for 10 cm^3^ of the skin or mucosa exceeded 14 Gy in one fraction due to a large tumor volume or if the target area was in close proximity to the OAR, the number of fractions was increased to avoid adverse effects such as skin ulcers. The dose–volume histograms of skin and mucosa were determined by contouring the areas of the skin and mucosa covered by the 14-Gy dose. The other dose limitations were as follows: 4 cm^3^ of the trachea or esophagus, 8 Gy; 0.35 cm^3^ of the spinal cord, 8 Gy. In this study, chemotherapy before/after CyberKnife was considered acceptable at the discretion of the attending physicians. Thirteen patients received chemotherapy before CyberKnife, including twelve patients who received concurrent chemoradiotherapy and one patient who received neoadjuvant chemotherapy before primary radiotherapy. None of the patients received chemotherapy after CyberKnife.

The median number of target LNs in the course of treatment was 1 (range, 1–7). Single-fraction radiotherapy was administered to 44 lesions (80%), whereas 3-fraction, 5-fraction, and 6-fraction radiotherapy were administered to 7, 3, and 1 lesion, respectively. The median prescribed isodose for the target was 64% (range, 49–89%). The median D95 was 20 Gy (range, 18–20 Gy) in 1 fraction (EQD2 = 50 Gy, BED10 = 60 Gy), 30 Gy (range, 27–33 Gy) in 3 fractions, and 31 Gy (range, 30–31 Gy) in 5 fractions. The median maximum dose (Dmax) was 31 Gy (range, 22–41 Gy) in 1 fraction (EQD2 = 106 Gy, BED10 = 127 Gy), 45 Gy (range, 44–57 Gy) in 3 fractions, and 48 Gy (range, 45–61 Gy) in 5 fractions.

### 2.3. Follow-Up Evaluations and Patient Data

Patients were followed up periodically. Diagnostic imaging techniques, including CT, MRI, and FDG-PET/CT, were performed to evaluate the tumor response every 2–3 months after CyberKnife treatment for at least the first 2 years. Toxicities were evaluated by physicians according to the Common Terminology Criteria for Adverse Events version 4.0.

### 2.4. Statistical Analysis

The OS and LC rates were calculated using the Kaplan–Meier method. The Log-rank test was used to compare the 1-year LC differences. Survival was measured in days from the initiation of the CyberKnife procedure to the time of death or final follow-up. Differences in the treatment effects according to LN volume and delivered dose were analyzed using the log-rank test. All statistical analyses were performed using Prism8 (GraphPad, San Diego, CA, USA).

### 2.5. Ethics Approval and Consent to Participate

This retrospective study was approved by our Institutional Review Board. This trial is registered in the University Hospital Medical Information Network Clinical Trials Registry (UMIN-CTR; number 000031155). Individual patient consent was waived due to the retrospective nature of this study.

## 3. Results

### 3.1. Tumor Response and LC after Treatment

The median follow-up period after CyberKnife treatment was 17 months (range, 2–53 months). The one-year LC and OS rates were 81.4% and 71.4%, respectively, and the two-year LC and OS rates were 81.4% and 46.3%, respectively (Figure 1). Local recurrence was observed in 10 lesions from seven patients. Two patients (12.5%) died from LN treatment failure, one (6.3%) died from skin ulcer bleeding, one (6.3%) died from primary site recurrence, one (6.3%) died from lung metastasis, and four (25%) died from other factors (pneumonia, two; subarachnoid hemorrhage, one; and comorbidity, one). The one-year LC was significantly higher with a target volume ≤1.0 cm^3^ than with a target volume >1.0 cm^3^; 95.8% versus 65.9%, respectively (*p* = 0.006) (Figure 2). There was no significant difference in the one-year LC when considering other factors, such as age, treatment interval, and the history of the previous surgery (Table 2).

### 3.2. Adverse Effects

The maximum grades of adverse effects during the follow-up period are presented in Table 3. Grade 2 pharyngitis was observed in two patients. Grade 1 dermatitis was observed in one patient, and grade 2 dermatitis was observed in one patient. Grade 1 anorexia was observed in one patient. With regard to the cutaneous region and mucosa, grade 3 or higher adverse effects were not observed. Fatal bleeding was observed in one patient who had a large tumor (91 cm^3^) with widespread invasion to the carotid artery and a cutaneous ulcer before CyberKnife treatment.

### 3.3. Dose–Volume Histogram

Sixteen patients were irradiated with a maximum dose of 14 Gy or more to the skin through CyberKnife treatment. The median of the maximum dose to skin was 23.0 Gy (range, 16.7–28.1 Gy), D 0.5 cc was 12.1 Gy (range, 8.1–18.2 Gy), and D 1 cc was 9.0 Gy (range, 3.6–14.5 Gy). Two patients were irradiated with a maximum dose of 14 Gy or more to the mucosa. The median of the maximum dose to mucosa was 28.0 Gy (range, 22.3–33.7 Gy), D 0.5 cc was 15.6 Gy (range, 4.2–27.1 Gy), and D 1 cc was 13.5 Gy (range, 3.0–23.9 Gy). Fourteen patients were irradiated with a maximum dose of 14 Gy or more to the carotid artery. The median of the maximum dose to the carotid artery was 21.9 Gy (range, 15.3–30.1 Gy), D 0.5 cc was 7.6 Gy (range, 0.4–15.4 Gy), and D 1 cc was 2.6 Gy (range, 0.0–12.3 Gy).

## 4. Discussion

The prescribed dose of re-irradiation has been shown to be an independent prognostic factor for LC, progression-free survival, and OS in cases of re-irradiation for locoregional recurrence or secondary primary cancer [28], as the authors reported that the three-year LC rate was 56% in patients receiving ≥58 Gy (EQD2), whereas the rate was 30% in those receiving <58 Gy (EQD2) (*p* < 0.05) with the use of conventionally fractionated radiotherapy for re-irradiation with concurrent chemotherapy. Regarding the prescribed dose to LNs, Wakatsuki et al. reported that the LC of untreated pelvic LNs associated with uterine cervical cancer significantly improved with a dose of ≥58 Gy (EQD2) [29]. The dose of 50 Gy (EQD2) for D95 in our study was lower than the ideal dose; however, the Dmax in the central part of the tumor was 1.56 times higher than the dose in the peripheral part of LN owing to a steep gradient dose distribution with CyberKnife (Figure 3 and Figure 4). The median Dmax was 106 Gy (EQD2) (range, 59–206 Gy), which might have contributed to the good LC. In the present study involving CyberKnife treatment for in-field neck LN recurrence, the two-year LC and OS rates were 81.4% and 46.3%, respectively. On the other hand, Yamazaki et al. reported the results of CyberKnife treatment for 107 cases of relapse, including those at the primary site and LN, within the head and neck irradiation field, using a median marginal dose of 30 Gy in 5 fractions (EQD2 = 40 Gy) [30]; the two-year LC and OS rates were 64% and 35%, respectively. Together, the better treatment outcome in our study than that in Yamazaki’s study may be attributable to the higher prescribed dose in our study [50 Gy (EQD2) for D95 in our study vs. 40 Gy (EQD2) for D95 in Yamazaki’s study], although results from the two studies cannot be compared directly without accounting for differences in study design and patient population.

The treatment of small LNs is easier than that of large LNs in terms of dosimetry. In the present study, 25 of 55 lesions had a small volume (≤1.0 cm^3^). Lesions with target volumes ≤1.0 cm^3^ had significantly better LC than lesions with target volumes >1.0 cm^3^, and the two-year LC rates were 95.8% and 65.9%, respectively (*p* = 0.006), suggesting the therapeutic advantage of small LNs even with re-irradiation. These results imply that the early detection of small neck LN recurrence using advanced modalities, such as high-resolution MRI and FDG-PET/CT with serum tumor markers, will help control in-field LN recurrence. In our study, two small LN of 0.05 cm^3^ in the same patient were included. One of them remained after chemotherapy and decided to be treated by CyberKnife. The other was close to the swelling LN (12.8 mm in diameter with the maximal standardized uptake value of 5.44). The distance between the small LN and the large LN was 3.6 mm so that the small LN was suspected as metastasis and decided to be treated by CyberKnife.

CyberKnife has the advantage of treatment with a small irradiation field [30] based on high set-up accuracy [28], which makes it possible to prescribe high doses to small targets and reduce the dose to the OARs. 

It is important to predict and prevent adverse effects associated with re-irradiation. A previous analysis has reported on the risk factors of internal carotid blowout syndrome (CBOS) after stereotactic radiotherapy (SRT) [31]. The authors mentioned that tumor invasion of the internal carotid artery more than half a round, presence of ulceration, and previous irradiation to areas of LNs were associated with internal carotid artery collapse. In our study, fatal collapse of the internal carotid artery was observed in one patient who had all those three factors 2 months after CyberKnife treatment. The patient received the dose of 36 Gy in 6 fractions and Dmax was 64 Gy (EQD2 = 48 Gy and Dmax EQD2 = 108 Gy, respectively). In this case, tumor invasion to the carotid artery was broad and the risk of blood vessel failure was estimated to be 75% at 6 months after CyberKnife [31]. The QOL of the patient was markedly impaired due to swelling, pain, and skin ulcer; therefore, the treatment performed according to the patient’s request for symptomatic relief, with sufficient informed consent.

Meanwhile, the safety and efficacy of salvage surgery for in-field neck LNs have not been established. Serious adverse events, including CBOS, after surgery occurred in three of eight patients who could resect LN relapse in the irradiation field [10]. While risk of surgery was high, the efficacy was limited; four out of eight patients had subsequence recurrence in the neck. Compared with previous reports about SRT and surgery, severe adverse events occurred less frequently in our study.

To avoid CBOS, the analysis for the optimal fractionation regimen is required. Yazici et al. adopted every other day SBRT protocol for re-irradiation [32]. They suggested that the protocol has a potential impact in terms of decreasing the incidence of fatal CBOS with maintaining the LC and overall survival (OS); CBOS-free median OS were 23 months and 9 months in every other day protocol and sequential protocol, respectively (*p* = 0.002). Therefore, the patients at high risk of CBOS may be worth examining every other day protocol.

All cases in our report were medically inoperable. In inoperable cases, knowledge about the effectiveness of immune checkpoint blockade has rapidly spread globally. The National Comprehensive Cancer Network guidelines version 3.2019 recommend nivolumab for patients with unresectable recurrent squamous cell head and neck cancer (category 1 recommendation) on the basis of high-quality evidence [33]. The one-year OS rate was found to be significantly higher with nivolumab than with standard chemotherapy (36.0% vs. 16.6%) [34]. Although nivolumab improves OS, the response rate is low, and it was found to be only 13.3% in the nivolumab group (six complete responses and 26 partial responses among 240 cases). In our study, CyberKnife treatment achieved a good tumor response rate, with two-year LC rate of 81.4% of the target LNs and two-year OS rate of 46.3% after treatment. Thus, the CyberKnife procedure could be a treatment option for oligo LN recurrence.

Recently, particle therapy including carbon-ion radiotherapy has emerged as the choice for re-irradiation. We showed the favorable results of carbon-ion radiotherapy for LN recurrence from gynecological cancers [35], in which carbon-ion radiotherapy showed three-year OS of 74% and no severe toxicity even after definitive radiotherapy. An in silico study reported that carbon-ion radiotherapy significantly reduced the mean dose to OAR compared to photons in re-irradiation of head and neck cancers [36]. Carbon-ion radiotherapy has a huge potential to overcome the in-field recurrences, even though the medical resources are limited.

The present study has several limitations. First, this was a retrospective study with a small number of patients and various primary disease sites. CyberKnife treatment has been increasingly used in Japan, whereas the incidence of in-field recurrence of head and neck tumors is approximately 15%; therefore, a longer study period was necessary to include a sufficient number of patients in the current study. Second, we were unable to analyze the details of previous chemotherapy and/or surgery because of the large heterogeneity in the reporting practices among institutions. Third, we did not perform biopsy for the pathological confirmation due to the potential side effects such as refractory skin ulcers and adopted CT, MRI, and FDG-PET/CT for the diagnosis of recurrence.

A future study including a greater number of patients is warranted to assess the advantages of CyberKnife treatment for LN recurrence in the irradiated field. Such investigations should include careful patient selection with consideration of tumor factors as well as patient history and characteristics in order to reduce severe adverse events.

## 5. Conclusions

Salvage treatment using the CyberKnife approach for in-field neck LN recurrence is safe and feasible in most patients, although it should be carefully considered for large and widespread tumors at the risk of fatal adverse effect including CBOS. Further study is needed to validate the advantages of CyberKnife as salvage treatment in larger cohorts.

## Figures and Tables

**Figure 1 jcm-08-01911-f001:**
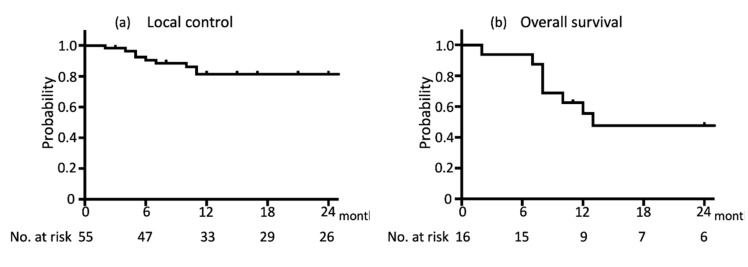
Kaplan–Meier survival estimates for (**a**) local control (*n* = 55) and (**b**) overall survival (*n* = 16) for CyberKnife-treated in-field neck lymph node (LN) lesions recurring after radiotherapy.

**Figure 2 jcm-08-01911-f002:**
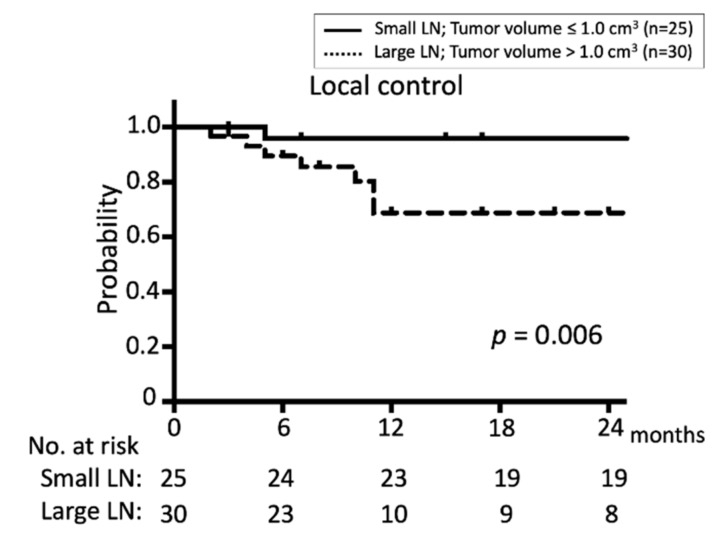
Kaplan–Meier survival estimates for local control or CyberKnife-treated in-field neck LN lesions recurring after radiotherapy stratified by target volume, i.e., ≤1.0 cm^3^ (*n* = 25) vs. >1.0 cm^3^ (*n* = 30).

**Figure 3 jcm-08-01911-f003:**
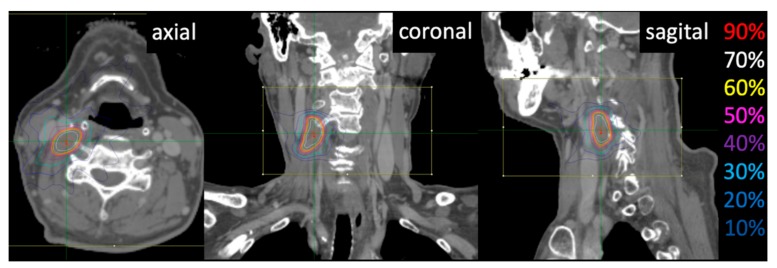
Dose distribution for CyberKnife in a representative case. A 58-year-old man with oropharyngeal cancer underwent radiotherapy with 60 Gy in 30 fractions. He showed neck lymph node recurrence after 12 months. He then received 30 Gy in 3 fractions by CyberKnife treatment. He is alive 20 months after CyberKnife treatment. Colored lines show 90% (red), 70% (white), 60% (yellow), 50% (pink), 40% (purple), 30% (cyan), 20% (blue), and 10% (dark blue) isodose curves.

**Figure 4 jcm-08-01911-f004:**
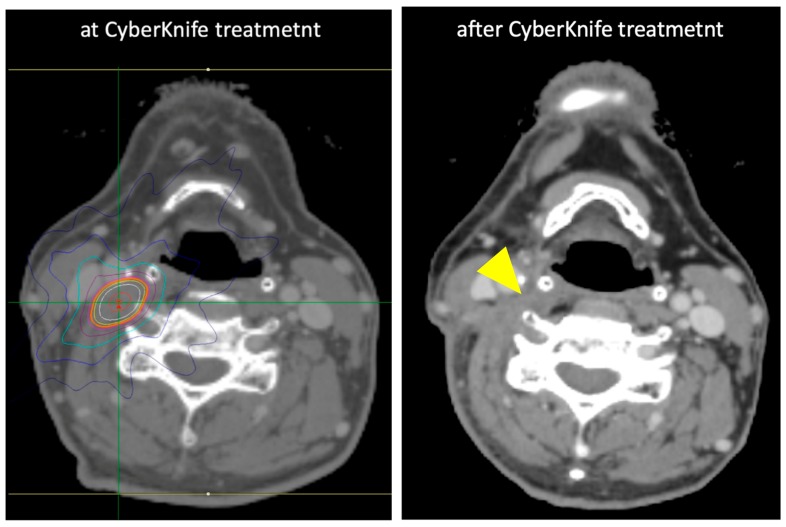
Treatment response after CyberKnife in a representative case. The case is as same as the case shown in Figure 3. Left figure shows the LN adenopathy at CyberKnife treatment. Right figure shows the complete response 20 months after treatment (arrowhead).

**Table 1 jcm-08-01911-t001:** Characteristics and treatment factors of patients with CyberKnife re-irradiation.

Variables	Strata	Patients (*n* = 16)	(%)	Lymph Node (*n* = 55)	(%)
Age		60 (45–72)			
Gender	Female	3	18.7	15	27.3
	Male	13	81.3	40	72.7
Primary site	Nasopharynx	3	18.7	17	30.9
	Oropharynx	3	18.7	3	5.5
	Hypopharynx	5	31.3	5	9.1
	Buccal mucosal	1	6.3	13	23.6
	Tongue	2	12.5	2	3.6
	Laryngeal	2	12.5	15	27.3
Surgical history	Yes	7	43.7	33	60
	No	9	56.3	22	40
Ulceration	Yes	3	18.7	4	7.3
	No	13	81.3	51	92.7
Tumor target volume (median)	(cm^3^)			1.2 (0.05–91)	
Prescribed dose of CyberKnife (median)	(Gy)			20 (18–36)	
Number of fractions (median)				1 (1–6)	
EQD2 (median)	[Gy (α/β = 10)]			50 (40–58)	
maximum dose (median)	(Gy)			31 (22–41)	
Treatment interval between primary RT and CyberKnife (median)	(months)			25 (6–69)	
Previous prescribed dose (median)	(Gy)			68 (50–70)	
Previous no. of fractions (median)				34 (25–35)	
Cumulative EQD2 (median)	[Gy (α/β = 10)]			116 (92–120)	

**Table 2 jcm-08-01911-t002:** Analysis of prognostic factors for local control rate after re-irradiation.

Variable	Strata	n	Median Follow-Up Period (Months)	1-Year LC	Hazard Ratio	*p*-Value
Age, years	≤60	29	41	78.5	1.5	0.51
	>60	26	11	85.5	0.6	
Gender	Male	40	15	77.0	1.7	0.43
	Female	15	41	92.9	0.6	
Primary site	Nasopharynx	17	41	81.3	NA	0.24
	Buccal mucosal	13	17	100		
	Laryngeal	15	37	85.1		
	others	10	9	50.6		
Previous surgery	Yes	33	21	89.1	0.2	0.059
	No	22	15	70.3	4.0	
Ulceration	Yes	4	6	80.7	0.3	0.54
	No	51	24	100	2.9	
Tumor volume	≤1.0 cm^3^	25	32	95.8	0.2	0.0061
	>1.0 cm^3^	30	10	65.9	5.8	
Prescribed dose of CyberKnife (EQD2)	<50 Gy	7	11	62.5	2.0	0.49
	≥50 Gy	48	23	83.8	0.5	
Treatment interval	≤25 months	28	32	79.6	1.7	0.41
	>25 months	27	15	83.5	0.6	

LC = local control rate, NA = not available, EQD2 = equivalent dose in 2-Gy fractions.

**Table 3 jcm-08-01911-t003:** Adverse events.

Adverse Events	Grade 1	Grade 2	Grade 3	Grade 4	Grade 5
pharyngitis	0	2	0	0	0
dermatitis	1	1	0	0	0
anorexia	1	0	0	0	0
injury to carotid artery	0	0	0	0	1

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
