# Peer review of "Efficacy and Feasibility of Salvage Re-Irradiation with CyberKnife for In-Field Neck Lymph Node Recurrence: A Retrospective Study"

_jcm, 2019, doi:10.3390/jcm8111911_

Round 1

Reviewer 1 Report

The authors wrote a manuscript about treatment of lymph node recurrences after primary radiotherapy with high dose re-irradiation using a CyberKnife. The authors addressed the importance that management of neck recurrences is challenging. Their study shows that using precise high dose radiotherapy gives promising results and very good local control.

There are some major concerns in this retrospective study. The gathered only sixteen patients in period of almost 8 years. To improve the statistics they took the number of recurrent lesions (n = 55), it questionable if this allow to do. It is the materials and methods section (lines 75 to 80) it is not clear how the recurrent lesions were defined exactly. No pathology confirmation of the lesions are mention, which is major drawback of the study. For one patient with a buccal mucosal tumor even had 13 recurrent lesions and two patient with a laryngeal tumor had 15 lesions together. Which seems to be a little bit unlikely and those 3 patients have a major impact on the statistics. For this reason no conclusion can be drawn from this study.

Some minor comments:

Line 87 is written: 2 (12%) received radiotherapy for local or LN recurrence after surgery. What exactly did these patients receive? What makes this different than the 4 patients receiving radiotherapy as adjuvant postoperative therapy? Line 90-91 is stated that 7 (5 + 2) patients received surgery of the neck. In Lines 85 to 87 it seems only be 6 (4 + 2). This does not seem to be correct? Line 99 is written: The treatment doses and fractions were decided according to the volumes of the tumor and 99 surrounding critical structures. This information is too limited. Please describe in more detail how treatment dose and fractions were decided. Line 115: Please explain in more detail how many patient received chemotherapy. Because influences OS en LC. Line 117: The range (1-7) of number of treated lesions does not seem to be correct. The patient of the buccal mucosa tumor did have 13 lesions. In Table 1 it is mentioned the interval between primary RT and CyberKnife is in a range of 1-69 months. This means that for one patient the interval was only 1 month. To my experience it is impossible to define after one month of initial radiotherapy if the lesion is residual of new (even with pathology confirmation). Please explain this patient in more detail. The authors mention in the last part of the discussion from line 258 briefly the limitations of their study. I miss why no pathology confirmation has been done and why the period of inclusion is long.

Reviewer 2 Report

LN recurrence is due to the presence of cancer stem cells as minimal residual disease  or circulating tumour cells, circulating cancer stem cells in the blood. Images of only at and after cyber knife is shown in this article. 

Requires revision to improve the sufficient sample size and images for justifying the cyber knife novelty. Group needs to be compared with out cyber knife treatments (with conventional RT and Chemotherapy). In order to assess the synergistic efficacy of cyberknife+chemo, images of  chemotherapy gp alone patients is required?

As indicated page 4 under the results section in line 140 the overall survival percentage survival was only 8 patient out of 16 patient combining cyberknife and chemotherapy group. Clear efficiency and potency should be measured with the correct control groups.

Round 2

Reviewer 1 Report

The authors adressed the questions and concerns well.

Although I think that the number of included patients is still a major drawback of this study.

It is goed enough to accept.

I still would advice the remake figure 3 and 4, the quality is poor.

Author Response

Comments and Suggestions for Authors

The authors adressed the questions and concerns well.

Response:

We sincerely extend our gratitude to the reviewer for evaluating our manuscript.

Q1: Although I think that the number of included patients is still a major drawback of this study. It is goed enough to accept. I still would advice the remake figure 3 and 4, the quality is poor.

Response:

We appreciate for your understanding.

To improve figure quality, we have clearly indicated targets in Figure 4. We hope these changes are sufficient answers to your suggestions.

Reviewer 2 Report

I agree now as technical correctness publication.

However, the title is longer, abstract concluding sentence and conclusions wordings are similar. These changes should be incorporated in the manuscript.

Author Response

Comments and Suggestions for Authors

I agree now as technical correctness publication.

Response:

We sincerely thank the reviewer for their evaluation of our manuscript. According to the reviewer's suggestions, we have made a revision on our manuscript as detailed in the following responses.

Q1: However, the title is longer, abstract concluding sentence and conclusions wordings are similar. These changes should be incorporated in the manuscript.

Response:

According to the reviewer’s comment, we changed the title and conclusions part as follows:

Title:

Efficacy and feasibility of salvage re-irradiation with CyberKnife for in-field neck lymph node recurrence: A retrospective study

Line 290

Salvage treatment using the CyberKnife approach for in-field neck LN recurrence is safe and feasible in most patients, although it should be carefully considered for large and widespread tumors at the risk of fatal adverse effect including CBOS. Further study is needed to validate the advantages of CyberKnife as salvage treatment in larger cohorts.
